# Switching Competitors Reduces Win-Stay but Not Lose-Shift Behaviour: The Role of Outcome-Action Association Strength on Reinforcement Learning

**Vincent Srihaput** [1], **Kaylee Craplewe** [2] **and Benjamin James Dyson** [1,2,3,*] 

1   School of Psychology, University of Sussex, Falmer BN1 9QH, UK; vinniesput@gmail.com
2   Department of Psychology, University of Alberta, Edmonton, AB T6G 2E9, Canada; craplewe@ualberta.ca
3   Department of Psychology, Ryerson University, Toronto, ON M5B 2K3, Canada
*   Correspondence: bjdyson@ualberta.ca

**Abstract:** Predictability is a hallmark of poor-quality decision-making during competition. One source of predictability is the strong association between current outcome and future action, as dictated by the reinforcement learning principles of win–stay and lose–shift. We tested the idea that predictability could be reduced during competition by weakening the associations between outcome and action. To do this, participants completed a competitive zero-sum game in which the opponent from the current trial was either replayed (opponent repeat) thereby strengthening the association, or, replaced (opponent change) by a different competitor thereby weakening the association. We observed that win–stay behavior was reduced during opponent change trials but lose–shiftbehavior remained reliably predictable. Consistent with the group data, the number of individuals who exhibited predictable behavior following wins decreased for opponent change relative to opponent repeat trials. Our data show that future actions are more under internal control following positive relative to negative outcomes, and that externally breaking the bonds between outcome and action via opponent association also allows us to become less prone to exploitation.

**Keywords:** win–stay; lose–shift; Rock-Paper-Scissors; mixed-strategy; associative learning

## 1. Introduction

During competition, we aim to exploit others and avoid exploitation. With imperfect or hidden information regarding one's opponent [1], the only way to guarantee the absence of exploitation in zero-sum games is to behave according to mixed-strategy (MS) [2–5]. Here, all actions must be equally likely in the long run and the selection of the next action must not be contingent on the outcome of the previous action. However, humans reliably deviate from MS due to the high cognitive load demanded by this strategy, and, contrary expectations about the correlated nature of outcome distribution in the real world [6,7]. These deviations (see [8], for a review) are often expressed via the operant conditioning (reinforcement learning) principles of win–stay and lose–shift [9]. If we are more likely to repeat an action following success or more likely to change an action following failure, then this increases both predictability and the chance of future exploitation.

Research from economics, evolutionary and cognitive psychology, all point to MS adherence being less likely following negative rather than positive outcomes. From an economic point-of-view, losses are over-weighted relative to gains even when both type of outcome have equivalent objective value [10]. From an evolutionary point-of-view, organisms cannot "afford to learn to avoid," given the potentially fatal consequences of failure [11,12]. Due to the increased impact of harm following loss relative to the impact of gain following win, it is reasonable that there is a reliable and automatic

response following loss in which the previous behavior is changed (i.e., lose–shift; [13,14]). From a cognitive point-of-view, failure states are naturally linked to the generation of negative affect that compromise the quality of future actions [15–18]. An unwillingness to remain in a failure state also leads to a reduction in decision-making time following loss (e.g., [19–24]). This can also give rise to more predictable responding, as seen in the increases in lose–shift behavior relative to win–stay behavior, compared against the values expected by MS performance (e.g., [25,26]). While there are also contexts in which win–stay behavior is more in evidence than lose–shift behavior (e.g., [27,28]), the present concern is the degree to which individuals can regulate the expression of win–stay and lose–shift, and what environmental features might trigger them into doing so.

For example, in manipulating the value of wins and losses in a zero-sum game, [29] showed differences in the flexibility of behavioral and neural responses following positive and negative outcomes. Specifically, the degree of win–stay (but not lose–shift) increased when differential values were assigned to wins and losses (e.g., +2 vs. −1, or, +1 vs. −2, both compared with a baseline condition of +1 vs. −1; see also [14]). Moreover, neural responses to outcome indexed by an early event-related potential (ERP) known as feedback-related negativity (FRN; [1]) fluctuate as a function of the value of wins but not losses (see also [30,31], in the context of positive and negative outcome probability manipulations). Therefore, the neural and cognitive state afforded by success allows for greater flexibility in the expression of win–stay behavior, relative to the neural and cognitive state afforded by failure and the expression of lose–shift behavior. Since the dynamics of competitive decision-making clearly contact with real-world domains such as problem gambling [32–34], it is important to consider how we can reduce the frequency with which individuals offer themselves up for exploitation. In the current paper, we test the ideas that a) breaking the associative bonds between current outcome and future action by deliberately switching (rather than repeating) opponents across consecutive trials should weaken traditional reinforcement learning rules, and, b) because of the flexibility of win–stay relative to lose–shift this reduction should be more apparent in the former rule than the latter rule.

Knowledge of opponent structure has previously been discussed in the context of one-shot games, where it is assumed that the participant will only have to interact with any given opponent once. Of note are the predictions that arise from normative models of decision-making during one-shot games in which, logically, participants should always defect in the Prisoner's Dilemma (e.g., [35]), and, should always offer the smallest proportion of money available to the other player in the Ultimatum game (e.g., [36]). Despite this, there is a surprising amount of 'economically irrational' and cooperative behavior exhibited in one-shot games [33]. While individuals in one-shot games behave more cautiously than predicted—as though they are actually treating the game as repeated and may have to encounter any given opponent again—there remain important distinctions between one-shot and repeated games. To quote [37], p. 190, italics in original: " . . . when people first think about repeated games, they often fall into the trap of assuming that any theoretical conclusions about a one-shot game can be applied to each repetition of it by the same players. The *supergame* that results when a stage game is repeated a number of times is, in fact, a new game with its own equilibrium points, and conclusions about the stage game cannot be applied straightforwardly to the supergame. Psychologically, however, players frequently think about each repetition as a separate game . . . ".

Thus, if participants adopt the stance that each round in a repeated game is independent as suggested by [37], this should weaken the associations between the outcome on the previous trial and the action of the current trial (such as win–stay, lose–shift). In contrast, if participants adopt the stance that each round is interdependent, this should strengthen these associations. As an explicit test of these two different positions, we introduced participants to multiple opponents across the same number of trials, but manipulated the organization of their appearance: half the opponents always reappeared on the very next trial representing proximal 'two-shot' encounters (interdependent; opponent repeat), and the other half always switched to another opponent on the very next trial representing distributed 'one-shot' encounters (independent; opponent switch). Under conditions

of opponent switch, we expect the associations between current outcome and future action to be weakened, thereby yielding less *win-stay* behavior[1].

## 2. Method

### 2.1. Participants

80 individuals completed the study across Experiments 1 and 2:40 (Experiment 1; 27 women, 36 right-handed; mean age = 20.85, sd = 1.44) at the University of Sussex under the School of Psychology approved ethics protocol ER/VS247/1, and, 40 (Experiment 2; 27 women, 36 right-handed; mean age = 20.85, sd = 1.44) at the University of Alberta under the Research Ethics Board 2 approved protocol PRO00086111. A £25 Amazon voucher was randomly allocated to one participant in Experiment 1; course credit was allocated in Experiment 2.

### 2.2. Stimuli and Apparatus

Eight cartoon-style avatars sourced from Freepik.com were used to represent each of the opponents (approximate on-screen size 4.5 cm × 4.5 cm). Pictures of two gloved hands representing the nine interactions between participant and opponent during Rock, Paper, Scissors (RPS) from [29] were also used (approximate on-screen size 10.5 cm × 4 cm). Stimulus presentation and response monitoring was conducted by Presentation (version 18.3, build 07.18.16). Participants also filled out a short version of the Cognitive Emotion Regulation Questionnaire (CERQ-short; [39]) at the end of the study, as part of a separate set of research questions.

### 2.3. Design

Participants completed 288 trials of Rock, Paper, Scissors (RPS) in a single block of trials[2]. Unbeknownst to the participants, the trials were organized into 144 pairs of trials presented in a random order, and only consecutive trials within those trial pairs were analyzed. Half (72) constituted opponent repeat pairs and the other half (72) constituted opponent change pairs. 4 opponents (O1, O2, O3, O4) were assigned to 18 repeat pairs each, where if an opponent appeared on the current trial $n$ then they also reappeared on the very next trial $n + 1$ (e.g., O1 to O1). 4 opponents (O5, O6, O7, O8) were assigned to 18 change pairs each, where if an opponent appeared on the current trial $n$ then one of the other opponent change avatars would appear on the future trial $n + 1$ (e.g., O5 to O6, O5 to O7, O5 to O8), with each of the three possible pairings occurring 6 times. This design ensured that all opponents appeared 36 times across the block. All opponents played MS, where 6 Rock, 6 Paper, 6 Scissors responses were randomized during both their allocated $n$ and $n + 1$ trials. Thus, in the long run, opponents would win–stay 33.3% and lose–shift 66.6% of the time. To add some variation in opponent performance in Experiment 1 (only), two opponents (one allocated to the repeat and one to the change condition) played in accordance with an item bias, playing 12 Rock, 3 Paper, 3 Scissors responses during both their allocated $n$ and $n + 1$ trials. Both repeat and change conditions consisted of two male and two female avatars, the assignment of avatar to group was counterbalanced, as was the assignment of male or female avatars to the item-biased opponents.

### 2.4. Procedure

Before beginning the study, participants were instructed that they would play RPS against eight different opponents, that some of the opponents may use different strategies, and, they were to

---

1　　Such predictions also appear in [38], p. 1081 in their suggestion that with opponent 'rotation,' participants "ought to feel freer to condition on their own past choices as an aid in achieving any desired move frequencies."

2　　For one participant, only 287 trials were recorded resulting in 143 pairs (286 trials) being used. For a second participant, only 285 trials were recorded resulting in 142 pairs (284 trials) being used.

try and get the highest score possible. There was never an indication that participants would be competing against other humans. At each trial, the participant's current score was displayed for 500 ms, after which the avatar of the current opponent appeared. This stayed on-screen until participants pressed 4, 5, or 6 on the keyboard corresponding to the selection of Rock, Paper, Scissors. This was replaced by the display of RPS selections for opponent (on the left; blue glove) and participant (on the right; white glove) for 1000 ms. This display was removed for 500 ms and then the outcome of the trial was presented for 1000 ms in the form of 'WIN' (+1; green font), 'LOSS' (−1; red font), or 'DRAW' (0; yellow font) as appropriate. The outcome was removed and the player's score was updated across a 500 ms period, after which the next trial began with the display of the next avatar. Participants were then thanked for their time and debriefed.

## 3. Results

At the group level [40,41] for Experiment 1, win–stay behavior was significantly reduced for opponent change trials relative to opponent repeat trials (41.88% versus 34.17%; $t[34] = 2.322$, $p = 0.026$) but the degree of lose–shift behavior did not change (81.83% versus 82.78%; $t[34] = 0.548$, $p = 0.586$). For Experiment 2, win–stay behavior was not significantly reduced for opponent change trials relative to opponent repeat trials (46.17% versus 42.24%; $t[34] = 1.350$, $p = 0.185$) and the degree of lose–shift behavior did not change (76.04% versus 76.63%; $t[34] = 0.795$, $p = 0.794$).

To formally compare performance across Experiments 1 and 2, two-way ANOVAs were completed separately for win–stay and lose–shift proportions with opponent (repeat, change) as a within-participants factor and Experiment (1, 2) as a between-participants factor. *Win-stay* behavior was significantly reduced for opponent change trials relative to opponent repeat trials (44.02% versus 38.21%; $F[2,78] = 6.951$, MSE = 0.019, $p = 0.010$, $\eta_p^2 = 0.082$). There was no main effect of experiment $F[1,78] = 2.802$, MSE = 0.054, $p = 0.098$, $\eta_p^2 = 0.035$), nor interaction $F[1,78] = 0.735$, MSE = 0.019, $p = 0.394$, $\eta_p^2 = 0.009$; see Figure 1) [3]. In contrast, lose–shift behavior was equivalent across opponent change and repeat trials (78.93% versus 79.70%; $F[1,78] = 0.292$, MSE = 0.008, $p = 0.591$, $\eta_p^2 = 0.004$), and did not interact with experiment $F[1,78] = 0.015$, MSE = 0.008, $p = 0.903$, $\eta_p^2 < 0.001$). A main effect of experiment $F[1,78] = 4.152$, MSE = 0.034, $p = 0.045$, $\eta_p^2 = 0.051$) indicated more lose–shift behavior in Experiment 1 relative to Experiment 2. This may have been due to the deployment of two opponents in Experiment 1 who had an item bias, with increases in participant win–stay and lose–shift behavior translating to increased win rates against these opponents. As predicted by previous data, the average lose–shift deviation from 66.6% was significantly larger than the average win–stay deviation from 33.3% (+12.65% versus +7.78%; $t[79] = 1.817$, $p = 0.037$, one-tailed).

As an additional test of increased mixed-strategy use following opponent change trials, binomial tests were carried out at the individual level [38] under the null hypothesis that their observed proportion of win–stay behavior was 33.3% and lose–shift behavior was 66.6%. Regarding win–stay behavior, the null could be rejected ($\alpha = 0.050$) for 38 individuals during opponent repeat trials versus 31 individuals during opponent change trials. Regarding lose–shift behavior, the null could be rejected for 44 individuals during opponent repeat trials versus 47 individuals during opponent change trials.

---

[3]    The decrease in win–stay behavior during opponent change relative to opponent repeat trials was replicated when the 2 opponents playing with an item bias were removed from the overall averages for Experiment 1 (43.55% versus 38.15%; $F[1,78] = 5.693$, MSE = 0.020, $p = 0.019$, $\eta_p^2 = 0.068$)

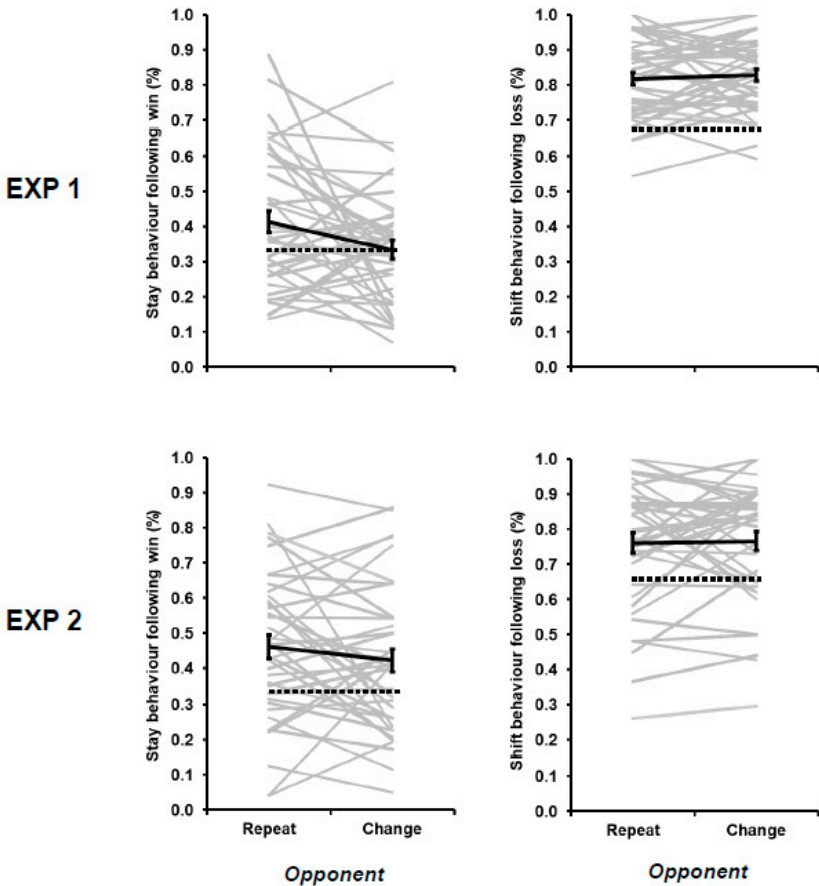

**Figure 1.** Individual (grey) and group (black) data for win–stay and lose–shift proportions across opponent repeat and opponent *change* trial pairs in Experiments 1 and 2. Error bars represent standard error. Dotted line represents proportion of win–stay (0.333) and lose–shift (0.666) behavior expected from unpredictable performance.

## 4. Discussion

By changing opponents across trials, the association between current outcome and future action can be weakened, thereby decreasing the degree to which individuals express predictable (and hence exploitable) behavior during competition. The caveat in our study is that reducing predictable behavior is more likely following positive rather than negative outcomes. First, participants deviated less from the value of win–stay relative to lose–shift behavior expected on the basis of unpredictable (random) performance. Second, win–stay behavior was decreased when the opponent changed between trials but lose–shift behavior did not change as a function of opponent replaying or replacing. Third, the number of individuals who were able to exhibit unpredictable behavior following wins increased when opponents changed [4]. These new observations are consistent with previous behavioral, neural, and, animal work showing that actions following negative outcomes are more impulsive relative to positive outcomes [42,43], that both behavioral and neural reactions following losses are less variable than those following wins [14,29–31], and that for the animal, lose–shift behavior is an "intrinsic feature of the brain's choice mechanisms that is engaged as a choice reflex" [13], p. 1.

---

[4] These differences in win–stay and lose–shift flexibility cannot be attributed to the variable experience of positive and negative outcomes in the current study. At a group level, the proportion of wins (33.51%) was equivalent to the proportion of losses (33.12%; $t[79] = 0.814$, $p = 0.418$). At an individual level, the degree of win–stay behavior was not correlated with individually experienced ($r = 0.106$, $p = 0.350$), nor was lose–shift behavior correlated with individually experienced lose rates ($r = -0.131$, $p = 0.247$).

The current study would appear to represent a relatively conservative test of the minimal requirements necessary to suppress reinforcement learning, given that participants performed (a) against computerized (rather than human) opponents, and, (b) in the absence (rather than presence) of financial incentive. In terms of the nature of opponency, participants have self-reported less presence, flow, and enjoyment playing computer rather than human opponents in online games [44], but are also more likely to tolerate inequitable offers from computer relative to human competitors [18]. One key difference between these two types of agent is that human players attempt to exploit one another in a tit-for-tat fashion, and this kind of 'dynamic coupling' [12,45] tends to be absent when computer opponents are deployed: in the current study, our program paid no attention to how our participants played nor did it attempt to exploit them [5]. Indeed, human participants are more likely to "stray from independence" when there is no threat of exploitation [41], p. 506. These observations also have echoes in the animal literature, where primates are more likely to behave in a vulnerable (exploitable) fashion if their opponent fails to take advantage of any behavioral bias [47]. So while the use of human opponents in future work might reduce the degree to which human participant offer themselves up to potential exploitation via the reinforcement learning rules of win–stay/lose–shift [48], this does not distract from our central result that these rules appear under different degree of control, given that participants down-regulated the use of win–stay but not lose–shift as a result of changing opponents between trials.

Another way in which motivation in the current study might have been relatively low as opposed to high, was in the absence of performance-related financial incentive. However, the impact of financial reward as an external contribution to competitive decision-making performance is neither reliable nor clear cut: Effects can be positive, negative or absent [49,50]. Experiments in our own laboratory directly comparing the presence or absence of financial reward in the context of zero-sum games similar to those reported here yield no significant effects [46], and, effects of incentive magnitude also tend to be weaker than other effects generated by decision complexity [49]. Performance-related pay can also complicate any mechanistic account of the data, since financial incentive as an external motivator can 'crowd out' or diminish the contribution of internal motivation [51]. Despite the use of computerized opponents and the absence of performance-related pay, zero-sum games are considered both intuitive and fun to play, thereby providing participants with some intrinsic motivation in a laboratory setting [48,52]. Since motivation remains an important individual difference, we are currently using the BIS/BAS scale [53] in our new work. This represents a particularly appropriate measure of motivation in the context of competitive decision-making, as the scales assesses behavioral inhibitory (BIS) and activation (BAS) systems responses to punishment and reward, respectively.

One question also remains regarding the extent to which the relative success of win–stay modulation and the relative failure of lose–shift modulation might be due to the initial weighting of these behaviors generated by the specific game context. For example, a typical pattern for performance against an MS opponent in a three-response zero-sum game such as Rock, Paper, Scissors is that the percentage of win–stay behavior is roughly equivalent to that predicted by MS (i.e., = 33.3%) whereas the proportion of lose–shift behavior tends to be larger than that predicted by MS (i.e., >66.6%; e.g., [26,27]). Alternatively, when, as in the present study, both win–stay and lose–shift deviate from expected MS values, lose–shift deviation is greater than win–stay deviation. This leads to the possibility that the use of 3+ response games naturally offers more response switch options (at least 2) relative to response repeat option (only 1). If participants are biased towards shift behavior solely as the number of response options increases, then using binary games contexts (e.g., [27,28]) should decrease any lose–shift bias and perhaps even reverse to a win–stay bias. While such accounts seem reasonable, there are counter examples in binary game contexts where lose–shift is dominant [14] and

---

[5] It is interesting to note that when comparing unexploitable with exploitable computer opponents, participants report an increased sense of co-presence for an automated program that played randomly [46].

in three-response game contexts where win-stay can be dominant [29]. Such an argument would also have to be predicated on the idea that more frequent behaviors are less likely to modulate. [28] propose the notion that 'surprise triggers change,' where a rare type of outcome might increase the degree of surprise elicited by that outcome, which in turn might encourage the organism towards behavioral modification. However, it would not seem this argument can be directly applied to the current context, in which the proportion of wins and losses was equivalent (see Footnote 4).

Finally, framing the data in terms of the ability to modulate win–stay and lose–shift mechanics would appear to be at odds with other models of zero-sum games such as those proposed by [12,45]. Here, ACT-R architecture (after [54]) is applied to item-based coding between consecutive trials (e.g., Rock-Paper-Paper), where a specific triad is nominated on the basis of strongest activation given the previous two trials, the triad is then analyzed to predict (and counter) the opponent's next move, and finally, the strength of triads are updated according to whether the counter action was successful or unsuccessful. This approach is attractive in preserving the limits of human working memory with respect to the representation of action history only two trials back (see also [55], in the context of visual search performance). Also, different versions of this architecture have helped to resolve models of human performance, where ambiguously-valenced draws tend to be interpreted more as negative rather than positive outcomes (see also [25,56,57]). Nevertheless, the storage of action triads (with strength representing the likelihood of initiation) continue to preserve basic notions of reinforcement and punishment, and one could imagine fundamental principles of operant conditioning being represented within this architecture on the basis of an opponent expressing a particular item bias such as Rock. In this particular example, win–stay could be represented by the temporary increased weighting of Paper-Paper-Paper, whereas lose–shift could be represented by the temporary increased weighting of Scissors-Rock-Paper [6]. Such ideas appear to fit well with the remit of the proposed ACT-R architecture: within competitive environments where opponents can be exploited, participants seek to maximize wins (rather than perform optimally; [45]). This may be somewhat different to the present cases where the majority of computerized opponents act in accordance with mixed-strategy, where wins cannot be maximized. Finally, it remains possible that ACT-R architecture could be overridden "under certain conditions" with "specific production rules" ([45] p. 114). Given the fundamental nature of operant conditioning, and the clear limitations of working memory in which associations between events are limited in size, it would seems reasonable that win–stay and lose–shift remain salient and in some cases automatic [13] reactions to the environment.

## 5. Conclusions

To conclude, the data make clear statements about when individuals have cognition under control and when their competitive behavior can be improved. By changing opponents and thus limiting the association between current outcome and future action, it is possible to reduce the predictability (and hence, exploitability) of individual stay behavior following wins. As such, the internal phasic state generated by positive outcomes along with the external experience of switching opponents makes it more likely that better performance will be expressed. In addition to extending time between trials [58], switching opponents can be added to the toolkit that helps to reduce the expression of typical reinforcement learning rules, thereby helping individuals who may be at their most cognitively vulnerable within competitive environments.

**Author Contributions:** V.S. and K.C. were responsible for investigation. V.S. and B.J.D. were responsible for writing the original draft. Correspondence should be addressed to: B.J.D. All authors have read and agreed to the published version of the manuscript.

---

[6]　More broadly, the mental representation of zero-sum games only with respect to the relationship between items can lead to identical behavior with more traditional representations that preserve both items and outcomes. For example, reference [8] discusses the behavioral isomorphism between zero-sum game strategy rules expressed in terms of the opponent's previous response only versus the participant's previous response and outcome.

**Funding:** The lab is funded by an NSERC Discovery Grant (RGPIN-2019-04954), an Alberta Gambling Research Institute Grant, and start-up monies provided by the University of Alberta (RES0042096).

**Conflicts of Interest:** The authors declare no conflicts of interest.

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
