# Peer review of "Switching Competitors Reduces Win-Stay but Not Lose-Shift Behaviour: The Role of Outcome-Action Association Strength on Reinforcement Learning"

_games, doi:10.3390/g11030025_

Round 1
Reviewer 1 Report
Manuscript Games-801647
Switching competitors reduces win-stay but not lose-shift behavior: The role of outcome-action association strength on reinforcement learning
Methodology:
The methodology is the weakest part of the paper. Whilst the paper’s argument is built on a suitable base of theory and ideas, the methodology used to get the data is poorly designed.
This paper is based on an experiment that uses the repetitive decisions of 80 individuals not motivated with appropriate incentives:
Incentives: One voucher randomly allocated to one participant in Experiment 1; course credit in Experiment 2 to each participant (flat payment not connected to participant´s decisions, effort or return)
Additionally, I am not sure I have understood he instructions. As the authors explain in p. 3, line 121, “there was never an indication that participants would be competing against other humans”.
Are they, actually? or not? Experiments about Ultimatum and Dictator game, for instance, show significant differences in behavior when playing against machines or against humans.
Are the methods employed appropriate to extract conclusions? I have serious concerns about the research design.
Reviewer 2 Report
This is an interesting manipulation but there are some issues. The experimental design and analysis are a bit odd. It was done as two separate experiments, in different labs,. But it is analyzed as if the two experiments were two conditions in the same experiment, presumably to get more power?? Foootnote 2 talks about how exp 1 would be significant if two subjects were thrown out. The results of the two experiments need to be reported separately, then a discussion about why and how to combine them can be had. Also, since there is a significant difference reported between the experiments, and the experiments had different procedures, it would be important to discuss how the difference could have caused the effect. Also, in exp 1 there was a manipulation on the behavior of the opponents but the results of this manipulation were not reported. Finally the authors seem to take the failure to get a significant effect as evidence of no effect, this needs to be corrected. These issues need to be fixed before the analysis and results can be properly evaluated.
A second problem is that the authors double down on one theoretical interpretation. There is nothing wrong with this interpretation but it is not the only possible one. The authors cite West and Lebiere but seem not to understand that West and Lebiere provide evidence for a different model. Specifically, they present evidence that humans play RPS using sequential dependencies, not move probability, and that they do this by using associative learning, not reinforcement learning. There are numerous papers by these authors supporting this model. This alternative account of human RPS should be reviewed and the results of the study should also be analyzed in terms of this model.
Round 2
Reviewer 2 Report
Changes look good